# A spectral-domain optical coherence tomographic analysis of *Rdh5*-/- mice retina

**Yuting Xie[1], Takayuki Gonome[1], Kodai Yamauchi[1], Natsuki Maeda-Monai[1], Reiko Tanabu[1], Sei-ichi Ishiguro[2], Mitsuru Nakazawa[1]***

**1** Department of Ophthalmology, Hirosaki University Graduate School of Medicine, Hirosaki, Japan,
**2** Department of Ophthalmology, Tohoku University Graduate School of Medicine, Sendai, Japan

* mitsuru@hirosaki-u.ac.jp

**Data Availability Statement:** All relevant data are within the manuscript and its Supporting Information files.

## Abstract

### Purpose

To investigate the longitudinal findings of spectral-domain optical coherence tomography (SD-OCT) in relation to the morphologic features in *Rdh5* knockout (*Rdh5*-/-) mice.

### Materials and methods

The mouse retina was segmented into four layers; the inner retinal (A), outer plexiform and outer nuclear (B), rod/cone (C), and retinal pigment epithelium (RPE)/choroid (D) layers. The thickness of each retinal layer of *Rdh5*-/- mice was longitudinally and quantitatively measured at six time points from postnatal months (PM) 1 to PM6 using SD-OCT. Age-matched C57BL/6J mice were employed as wild-type controls. The data were statistically compared using Student's *t*-test. The fundus appearance was assessed, histologic and ultrastructural examinations were performed in both groups.

### Results

Layers A and B were significantly thinner in the *Rdh5*-/- mice than in the wild-type C57BL/6J mice during the observation periods. Layers C and D became thinner in the *Rdh5*-/- mice than in the wild-type mice after PM6. Although no abnormalities corresponding to whitish fundus dots were detected by SD-OCT or histologic examinations, the intracellular accumulation of low-density vacuoles was noted in the RPE of the *Rdh5*-/- mice by electron microscopy. The photoreceptor nuclei appeared less dense in the *Rdh5*-/- mice than in the wild-type mice.

### Discussion

The results from the present study suggest that although it is difficult to detect qualitative abnormalities, SD-OCT can detect quantitative changes in photoreceptors even in the early stage of retinal degeneration induced by the *Rdh5* gene mutation in mice.

**Funding:** MN Grant-in-Aid for Scientific Research, C-19K09926 and C-16K11313 from Japan Society for the Promotion of Science (Tokyo, Japan) N M-M Grant-in-Aid for Early Career Scientists, B-17K16954 from Japan Society for the Promotion of Science (Tokyo, Japan).

**Competing interests:** The authors have declared that no competing interests exist.

# Introduction

Fundus albipunctatus (FA), as a type of hereditary retinal dystrophy, is a rare eye disorder characterized by an impaired visual ability under dim-light conditions and the presence of numerous white dots that are especially abundant near the mid-periphery and perifovea of the retina [1, 2]. It is inherited in an autosomal recessive pattern. Although mutations of the retinaldehyde binding protein1 (*RLBP1*) and retinal pigment epithelium (RPE)-specific-65-kDa protein (*RPE65*) genes have been reported to be associated with FA [3, 4], most cases of FA have been caused by mutations in the 11-cis retinol dehydrogenase 5 (*Rdh5*) gene [5, 6].

The *Rdh5* gene encodes the 11-cis retinol dehydrogenase 5 (RDH5), which is predominantly expressed in the retinal pigment epithelium (RPE), where it converts the molecule 11-cis retinol to 11-cis retinal [7–10]. 11-cis retinal is the recycling molecule in an integral operation of the visual cycle. It is the chromophore residing in rhodopsin and cone opsins and is needed for the conversion of light to electrical signals [11, 12].

FA as a form of congenital night blindness was originally considered a stationary disease. The scotopic electroretinography (ERG) amplitudes were reportedly significantly reduced when recorded after a standard period of dark adaptation but returned to the normal range after prolonged dark adaptation [1, 6, 13]. Yamamoto et al. suggested that the RPE produces 11-cis retinal at a slower-than-normal rate in genetically confirmed patients with *Rdh5* mutations, which may be interpreted by the slow recovery of light sensitivity in ERG after prolonged dark adaptation [5]. Subsequently, Yamamoto et al. reported that mutations in the *Rdh5* gene may play a role in the devolvement of macular atrophy with fading of the white dots in FA [14]. Other researchers also noted that the white dots disappeared or became weakened in patients with *Rdh5* mutations with increasing age or after uveitis [2, 15].

It was speculated that mutations in the *Rdh5* gene may play an important role in the progression of the white dots in the natural course of FA. However, *Rdh5* knockout (*Rdh5*-/-) mice reportedly do not show the typical white dots and retinal degeneration as in humans. Instead, *Rdh5*-/- mice show normal dark adaptation under standard conditions but display delayed dark adaptation under intense lighting [16–18]. In addition, no abnormalities have been detected in the structure or function of the retina of the *Rdh5*-/- mice, possibly because these mice have other compensatory pathways for regenerating chromophores [19].

Recent studies have shown that FA is not always stationary. Using spectral-domain optical coherence tomography (SD-OCT), researchers have revealed details, such as structural abnormalities in the outer retina and disruption of the junction of the photoreceptor, in some patients with FA [2, 20]. Although SD-OCT is a noninvasive technology that allows us to investigate the morphological characteristics of various retinal diseases [20, 21], it is impossible to directly evaluate the pathological features. In contrast, animal models associated with known gene mutations provide strong evidence that facilitate our understanding of the relationship between the SD-OCT findings and the pathological background [22–32]. However, to our knowledge, no study has assessed the relationship between SD-OCT findings and the origin of the structural changes in *Rdh5*-/- mice.

In this study, we explored the relationship between the SD-OCT findings and the morphologic features caused by mutations in the *Rdh5* gene in mice. We believe that the information obtained from this study will help clarify the relationship between the SD-OCT images of FA and the pathologic features in clinical practice.

## Materials and methods

### Experimental animals

This study was carried out in strict accordance with the regulations of the Association for Research in Vision and Ophthalmology (ARVO) Statement for the Use of Animals in Ophthalmic and Vision Research. The protocol was approved by the Committee on the Ethics of Animal Experiments of the Hirosaki University (Approval Number: M12023). All procedures were performed under general anesthesia as described below, and all efforts were made to minimize suffering.

The *Rdh5*−/− (B6;129S-Rdh5tm1Drie/J) mice had been created using C57BL/6J (B6) as host mice and 129/Sv (129S) as donor strain by Driessen, et al. [16] and were kindly provided by Dr. Mathew M. LaVail (University of California, San Francisco, CA, USA). C57BL/6J mice were purchased from Clea, Japan (Tokyo, Japan) and were used as wild-type controls. The mice were maintained at the Hirosaki University Graduate School of Medicine Animal Care Service Facility under a cycle of 12 h of light (50 lx illumination) and 12 h of darkness (<10 lx environmental illumination) in an air-conditioned atmosphere. Mice were given free access to food and water.

### SD-OCT examination and fundus photography

SD-OCT and fundus photography were performed by the methods as described in detail previously, using a Micron® IV, Image-Guided 830 nm OCT (Phoenix Retinal Imaging System, Phoenix Research Labs, Pleasanton, CA, USA) [22, 23, 25, 27]. In brief, SD-OCT and fundus photography were carried out at 6 points of time from postnatal-month (PM) 1 to PM6 for both *Rdh5*−/− and C57BL/6J mice. Four to eight mice were investigated at each time point. The mice were anesthetized with an intraperitoneal injection of a mixture of medetomidine hydrochloride (0.315mg/kg), midazolam (2.0mg/kg), and butorphanol tartrate (2.5mg/kg). The pupils were dilated with the instillation of eye drops containing a mixture of 0.5% tropicamide and 0.5% phenylephrine hydrochloride. The mouse ocular fundus was simultaneously monitored by a fundus camera equipped to the Micron® IV, and the position of the retinal SD-OCT image was set circumferentially around the optic disc (360˚; diameter, 500 μm; 140 μm away from the optic disc margin, Fig 1) [22]. To analyze the structure of a certain fundus change of interest, the position of the SD-OCT image was set vertically or horizontally, depending on the finding. The corneal surface was protected using a 1.5% hydroxyethylcellulose solution. Fifty images were averaged to eliminate projection artifacts. During all experimental procedures, the physical condition of the mice was monitored by inspection and palpation by the researchers.

### The analysis of the retinal layer thickness

The retina and choroid were divided into four layers, i.e. the inner retina (A), outer retina (B), photoreceptor segments (C), and RPE/choroid (D) layers. The inner retinal layer A consisted of the retinal nerve fiber layer (NFL), the ganglion cell layer (GCL), the inner plexiform layer (IPL) and the inner nuclear layer (INL); the outer retinal layer B consisted of the outer plexiform layer (OPL) and the outer nuclear layer (ONL); the photoreceptor segments layer C consisted of the photoreceptor inner segment (IS) and outer segment (OS) layers; and combined RPE and the choroid layer (D) (S1 Fig). As previously reported [23, 25, 27], we measured the thickness of layers A, B, C and D using the InSight® software program (Phoenix Research Labs). The borderline between each layer was automatically identified by the software program using the SD-OCT images and was manually modified by the researchers when necessary. The averages were obtained from both eyes of the same animal. The overall average retinal layer

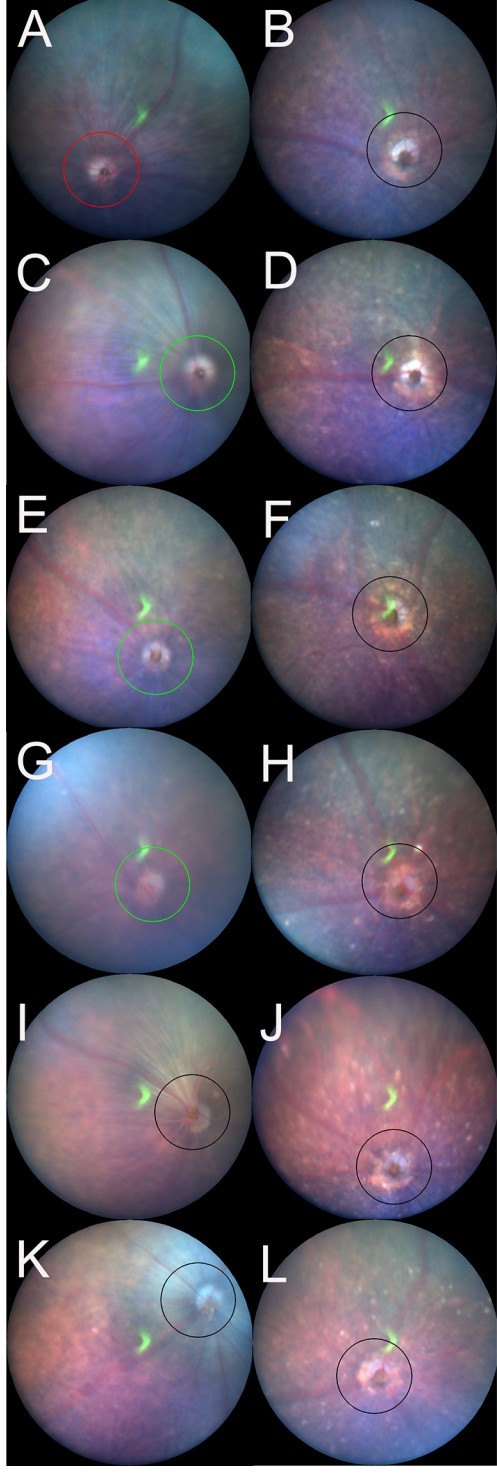

**Fig 1. Representative fundus pictures of the C57BL/6J (A, C, E, G, I, and K) and *Rdh5*<sup>-/-</sup> (B, D, F, H, J, and L) mice.**
Panels A, C, E, G, I, and K correspond to the fundus findings at postnatal-month (PM) 1, PM2, PM3, PM4, PM5, and PM6 of C57BL/6J mice, respectively. Panels B, D, F, H, J, and L correspond to the fundus findings at PM1, PM2, PM3, PM4, PM5, and PM6 of *Rdh5*<sup>-/-</sup> mice, respectively. Circles indicate the line at which the OCT images were created.

thickness was presented as the mean ± standard deviation after the normality of each distribution was confirmed by the Shapiro-Wilk test and Kolmogorov-Smirnov test.

## Histological examinations

Histological examinations were performed using eyes enucleated from both *Rdh5*<sup>-/-</sup> and C57BL/6J mice in PM3, PM4 and PM6 by the method previously described [22]. Immediately after euthanasia by luxation of the cervical spine, the eyes were excised under a microscope. To prevent possible artificial retinal detachment during further processing, an aliquot of 2% glutaraldehyde and 2% paraformaldehyde solution (pH 7.4) was injected into the anterior chamber through the corneal limbus. After fixation in the same solution for 2 h at room temperature, the eyeballs were re-fixed in 4% paraformaldehyde solution (pH 7.0) for 24 h at 4˚C. Paraffin embedding, sectioning, and hematoxylin and eosin (HE) staining were performed as previously described [22]. The HE-stained sections were photographed under a light microscope (DP-74, Olympus, Tokyo, Japan). The histological findings were compared to the corresponding findings of the SD-OCT images.

## Electron microscopy

Electron microscopy was performed using eyes enucleated from both *Rdh5*<sup>-/-</sup> and C57BL/6J mice in PM4 and PM6 according to the method previously described [22]. Similarly to the preparation for the histological examination, after enucleation, the eyes were immediately fixed with 2.5% glutaraldehyde and 2% paraformaldehyde solution (pH 7.4) for 24 h at 4˚C. An aliquot of the same fixation solution was injected into the anterior chamber. The retina and choroid were dissected out, post-fixed in phosphate-buffered 1% osmium tetroxide (pH 7.4) for 3 h at 4˚C, dehydrated in an ascending ethanol series (50%-100%), and embedded in epoxy resin. Thin sections (80–90 nm) were stained in uranyl and lead salt solutions. The sections were photographed by a transmission electron microscope (H-7600; Hitachi, Tokyo, Japan) at 100 kV.

## Statistical analysis

The statistical analyses of the data obtained in the present study were performed using the SPSS software program (version 25; Statistical Package for the Social Sciences, Chicago, IL, USA). The segmentation data from the two groups were compared using a two-way repeated analysis of variance (ANOVA) after the normality and equality of each distribution were confirmed by the Shapiro-Wilk test and Kolmogorov-Smirnov test, respectively. Student's *t*-test was performed to analyze differences in OCT segmentation between age-matched mice groups. *P* values of $< 0.05$ were considered as statistically significant.

## Results

### The fundus findings of C57BL/6J and *Rdh5*<sup>-/-</sup> mice

The longitudinal fundus changes of both C57BL/6J and *Rdh5*<sup>-/-</sup> mice are shown in Fig 1. In the *Rdh5*<sup>-/-</sup> mouse fundi, diffuse whitish spots appeared, especially after PM2 (Fig 1D, 1F, 1H, 1J and 1L), that were not be noted in the C57BL/6J mice (Fig 1C, 1E, 1G, 1I and 1K). However, the presence of such spots has not been previously reported in *Rdh5*<sup>-/-</sup> mice [16].

## The qualitative analyses of the SD-OCT findings in relation to the retinal structure in *Rdh5*$^{-/-}$ mice

To investigate the characteristics of the retinal development of *Rdh5*$^{-/-}$ mice, both C57BL/6J and *Rdh5*$^{-/-}$ mice were subjected to SD-OCT. The typical long-term OCT findings of C57BL/6J and *Rdh5*$^{-/-}$ mice are presented in Fig 2. In SD-OCT images from both *Rdh5*$^{-/-}$ and C57BL/6J mice, the retinal layers appeared to be consistent throughout the observation periods. No qualitative differences were observed in the retina between *Rdh5*$^{-/-}$ and C57BL/6J mice (Fig 2). In addition, this tendency was consistent even in the mid-peripheral areas (Fig 3B and 3D). The wavelike features in some OCT images (Fig 2) were artificially created dependent on the angle between the mouse eye and the eye lens attached to the apparatus. Therefore, this phenomenon was not due to any of phenotypic characteristics but was based on properties of the device.

Although we tried to identify the location of the whitish spots in the *Rdh5*$^{-/-}$ mice fundi by SD-OCT, we were unable to detect any corresponding lesions or hyper-reflective elements on SD-OCT images (Fig 3). This point differs from the findings of a previous report which described that discrete hyperreflective spots were reportedly detected in the retina of patients with FA associated with mutations in the *Rdh5* gene [33].

## The quantitative analyses of the OCT findings in the *Rdh5*$^{-/-}$ mice

The longitudinal changes of the thickness of the retinal layer are shown in Fig 4, S1 and S2 Tables. Statistically significant differences were found in the thickness of the layers A, B, C and D between two groups at different time points. The thickness of the layer B of the *Rdh5*$^{-/-}$ mice revealed thinner at all time points in comparison to those of the C57BL/6J mice (Fig 4B). The thickness of layer A showed similar tendency to that of layer B (Fig 4A). In the *Rdh5*$^{-/-}$ mice, retinal layers C and D became significantly thinner compared to those of the C57BL/6J mice in the late stage (Fig 4C and 4D). These quantitative differences between *Rdh5*$^{-/-}$ and C57BL/6J mice have not been described previously.

## Retinal morphology by HE staining and electron microscopy

The retinal morphology of the *Rdh5*$^{-/-}$ and C57BL/6J mice was examined by both light and electron microscopy. Histologically, all cell layers of the retina were comparable between the two groups, with no qualitative differences noted, although the width of the ONL appeared to be thinner in *Rdh5*$^{-/-}$ mice than in C57BL/6J mice (Fig 5). On electron microscopy, the accumulation of low-density vacuoles was detected in the RPE cells of the *Rdh5*$^{-/-}$ mice (Fig 6). As was detected by SD-OCT (Fig 4B), the width of the ONL in *Rdh5*$^{-/-}$ mice was narrower than in C57BL/6J mice (Fig 7). In addition, the interspaces between the nuclei of the photoreceptors in the ONL of *Rdh5*$^{-/-}$ mice were wider than in C57BL/6J mice (Fig 7). Because the intracellular organelles were observed in the space, these spaces were considered to be cytoplasm of photoreceptor and/or Müller cells. No qualitative differences were found in the retinal layers A and C between C57BL/6J and *Rdh5*$^{-/-}$ mice.

## Discussion

In this study, we first described the SD-OCT findings of the retina of *Rdh5*$^{-/-}$ mice in relation to the morphologic findings. The results obtained may expand our knowledge of the interpretation of the SD-OCT findings in *Rdh5*$^{-/-}$ mice and may provide some clues for future studies attempting to further our understanding of FA.

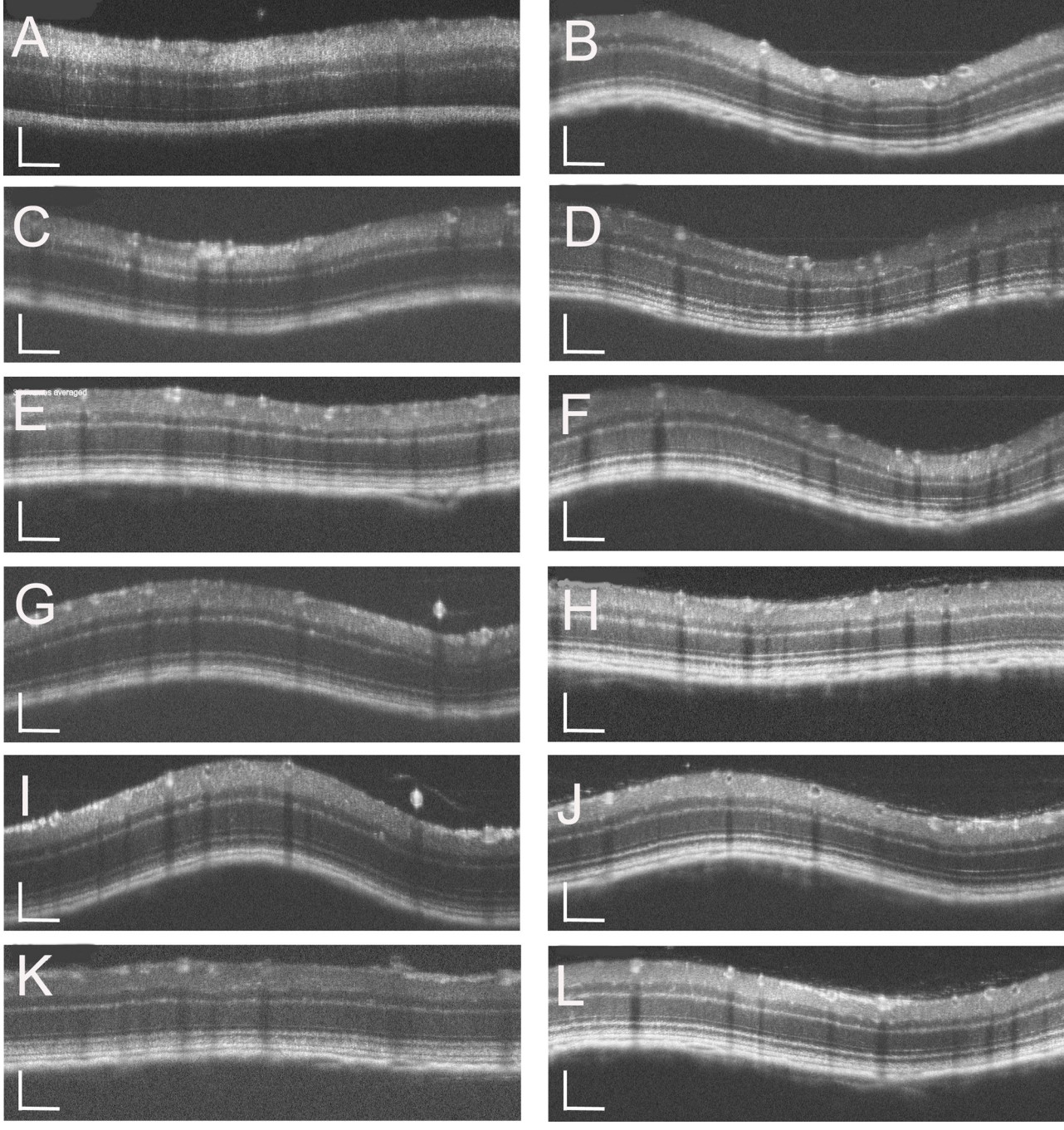

**Fig 2. Representative OCT images of the C57BL/6J (A, C, E, G, I, and K) and *Rdh5*$^{-/-}$ (B, D, F, H, J, and I) mice.** Panels A, C, E, G, I, and K correspond to PM1, PM2, PM3, PM4, PM5, and PM6 of C57BL/6J mice, respectively. Panels B, D, F, H, J, and L correspond to PM1, PM2, PM3, PM4, PM5, and PM6 of *Rdh5*$^{-/-}$ mice, respectively. The position of the retinal SD-OCT image was set circumferentially around the optic disc (360˚; diameter, 500 µm; 140 µm away from the optic disc margin, indicated by circles in Fig 1). The wavelike features in some images were artificially created depended on the angle of the mouse eye against the eye lens. The SD-OCT images were created using light stimulation at 830 nm. Bar indicates 100µm.

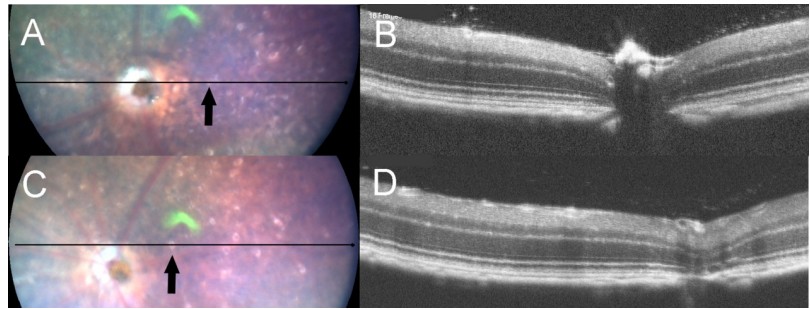

**Fig 3. The fundus photographs (panels A and C) and corresponding OCT images (panels B and D) of *Rdh5*<sup>-/-</sup> mice.** Panels A and B were taken at PM4, and panels C and D were recorded at PM6. The lines in the left panels A and C indicate the section line of the corresponding OCT images in the right panels B and D, respectively. Note that the orientations of B and D are opposite to those of A and C. Arrows indicate the location of white spots.

**Fig 4. The longitudinal changes in the thicknesses of the retinal layers.** Open circles, C57BL/6J mice; closed circles, *Rdh5*<sup>-/-</sup> mice. Panel A, thickness changes in the inner retinal layer (layer A). Panel B, the thickness of the outer retinal layer (layer B). Panel C, the thickness of the photoreceptor inner and outer segments (IS/OS) layer (layer C). Panel D, the thickness of the combined RPE and choroid layers (layer D). Animal numbers: C57BL/6J mice, n = 4 (PM1-4), n = 6 (PM5 and 6); *Rdh5*<sup>-/-</sup> mice, n = 6 (PM1-3 and PM6), n = 8 (PM4), n = 4 (PM5). Statistical significance: *, $P < 0.05$; **, $P < 0.01$; ***, $P < 0.001$ (Student's *t*-test).

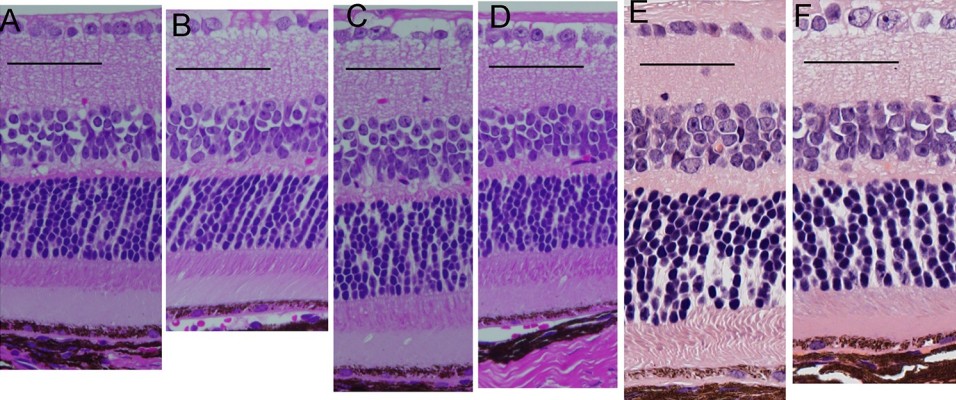

**Fig 5. Light microscopic findings by hematoxylin and eosin staining.** Retinal sections of C57BL/6J (A) and *Rdh5*<sup>-/-</sup> (B) mice at PM3, C57BL/6J (C) and *Rdh5*<sup>-/-</sup> (D) mice at PM4, and C57BL/6J (E) and *Rdh5*<sup>-/-</sup> (F) mice at PM6, respectively.

Our *Rdh5*<sup>-/-</sup> mice displayed whitish spots after PM2 while the C57BL/6J mice showed normal fundus throughout the time course (Fig 1). FA associated with mutations of the *Rdh5* gene in human is characterized by the deposition of numerous punctate white spots in the retina [5, 6]. It has been reported that these spots exist even on the day of birth and show little change afterward [33]. Conversely, it has also been reported that the white spots that appeared in the mid- and far-periphery tend to be convergent and shrink, becoming even less apparent with age [2, 12, 15]. The whitish spots on the eyes of our *Rdh5*<sup>-/-</sup> mice gradually increased after PM2 and until PM6, these spots showed no tendency to become less obvious or transparent. This may be due to species-related differences between mice and humans. SD-OCT in patients with FA has shown hyper-reflective elements corresponding to white dots in ocular fundus [2, 34]. Querques et al. reported that the disruption of the photoreceptor layer in a patient with FA associated with cone dystrophy at the level of the OS [20]. Shatz et al. and Sergouniotis et al. revealed that hyper-reflective lesions had extended from the RPE to the external limiting membrane [34, 35]. However, we found no such phenomenon in the fundi of the *Rdh5*<sup>-/-</sup> mice.

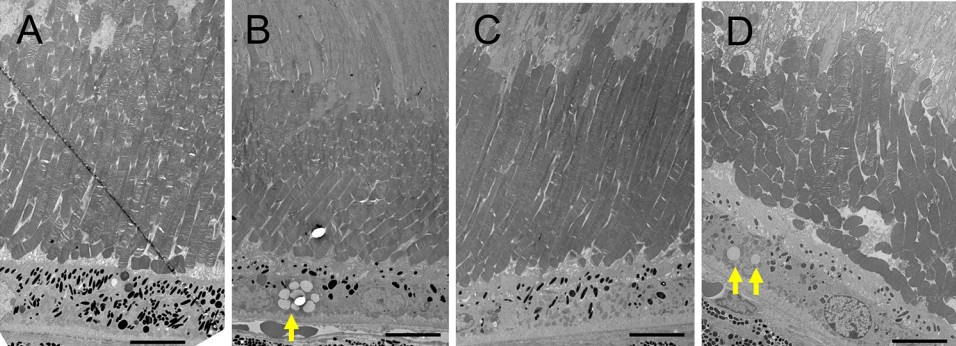

**Fig 6. Electron microscopic findings of the photoreceptor inner and outer segment layer and the retinal pigment epithelial cells (RPE) of C57BL/6J (A) and *Rdh5*<sup>-/-</sup> (B) mice at PM4, and C57BL/6J (C) and *Rdh5*<sup>-/-</sup> (D) mice at PM6, respectively.** The arrows indicate low electron-density vacuoles in the RPE of *Rdh5*<sup>-/-</sup> mice. Bars indicate 10 μm.

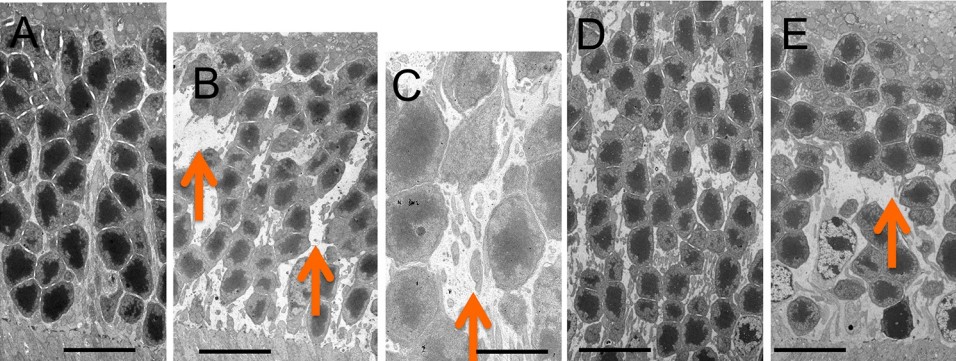

**Fig 7. Electron microscopic findings of the outer nuclear layer of C57BL/6J (A) and *Rdh5⁻/⁻* (B and C, respectively) mice at PM4, and C57BL/6J (D) and *Rdh5⁻/⁻* (E) mice at PM6, respectively.** Arrows indicate wide interspaces between nuclei of the photoreceptors. Bars indicate 10 μm in panels A, B, D and E, and 2 μm in panel C.

The pathology of FA can be related to the accumulation of cis-retinyl esters [16, 36], and similarly, the accumulation of 13-cis-retinyl ester in RPE can be suspected to be involved in *Rdh5⁻/⁻* mice [16, 17, 37], possibly leading to formation of the small whitish dots seen in the present study. The lack of RDH5 in the *Rdh5⁻/⁻* mice can be countered by high concentrations of 11-cis-retinol [36]. Mice are able to adjust or make use of different metabolic pathways for retinoid metabolism more efficiently than humans, resulting in less noticeable retinal pathology [16, 19]. According to Kim et al., no discernible differences were observed in 2-month-old *Rdh5⁻/⁻* mice retina based on findings of transmission electron microscopy [19]. In mice, although RDH5 is the principle enzyme that produces 11-cis-retinal in the RPE, RDH11 and RDH10 play minor but complementary roles in 11-cis-retinal regeneration [17–19, 36, 38, 39]. The results from these previous studies suggest that 11-cis-retinal production may also be catalyzed by other enzymes than RDH5 in the mouse RPE. This may explain why it was difficult to detect any changes in 2-month-old mice retina by electron microscopy in the previous study [19]. In this study, we found the low-density vacuoles accumulated in the RPE of the *Rdh5⁻/⁻* mice at PM4 and PM6 (Fig 6), suggesting that these vacuoles may be derived from the accumulated cis-retinoids and cis-retinyl esters [16].

However, electron microscopy showed 1) the accumulation of low-density vacuoles in the RPE of *Rdh5⁻/⁻* mice and 2) the reduction in the photoreceptor cell density and abundance of the cytosolic space in the ONL. The photoreceptor cell density was relatively scarce in the *Rdh5⁻/⁻* mice compared to C57BL/6J mice (Fig 7). Although the exact mechanism is still unclear, this electron microscopic finding may explain the thinning of the retinal layer B (OPL and ONL) observed by SD-OCT in *Rdh5⁻/⁻* mice (Fig 4) and the thinning of the ONL suggested by histologic analyses (Fig 5). This phenomenon indicates an advantage of SD-OCT over other examinations, as it can detect quantitative changes in the retina more sensitively than a histologic study. In addition, these structural changes detected by SD-OCT may be the earliest abnormality of the *Rdh5⁻/⁻* mice retina, in which overall retinal photoreceptor photoexcitation function is still maintained. Actually, this tendency was observed previous report [16, in their Fig 3F]. However, it is possible that these quantitative changes observed by SD-OCT may be related to the defective photo-recovery function in *RDH5⁻/⁻* mice [16, 19]. Further studies should be carried out in the future to clarify this point. Although SD-OCT was unable to detect any abnormal lesions corresponding to the numerous fundus whitish spots, the abnormal accumulation of low-density vacuoles (Fig 6B) may be related to the pathogenesis of the

fundus whitish dots. Based on their configuration and color tones, these dots may have been atrophic changes in the RPE [40] and appear to be difficult to be detected by SD-OCT. While the mechanism underlying why layer A was thinner in *Rdh5$^{-/-}$* mice than in C57BL/6J on SD-OCT remains unclear, it is possible that this phenomenon is secondary to the deficiency of RDH5. However, further studies will be needed in order to clarify this point.

Patients with FA may have dysfunction of the cone and rod systems throughout the retina [39, 41–43]. Although we did not note any morphological abnormalities of the IS or OS layers in our *Rdh5$^{-/-}$* mice, the photoreceptor IS/OS layer became thinner in observational periods after PM4 (Fig 4C). Likewise, the RPE/choroid layer also became thinner in the late stage (Fig 4D). These results suggest that the retinal changes seen in the present study may be those in the early stage of retinal degeneration caused by defect in the *Rdh5* gene in mice.

In conclusion, although the SD-OCT technology at present has limited utility in differentiating degeneration and the size variability of the retina, it has the advantages of noninvasively observing longitudinal quantitative changes in the thickness of each retinal layer more sensitively than a histologic study. These results suggest that SD-OCT can detect quantitative abnormalities in photoreceptors induced by *Rdh5* gene mutations even in the early stage of retinal degeneration, which may help clarify the molecular pathogenesis of FA.

## Supporting information

**S1 Fig. Comparison between SD-OCT layers and histological features.** Definition of retinal sublayers A, B, C and D, ELM, IS-EZ and IZ, and comparison between a representative OCT image and histological findings of an *Rdh5$^{-/-}$* mouse at PM3. Abbreviations: NFL, nerve fiber layer; GCL, ganglion cell layer; IPL, inner plexiform layer; INL, inner nuclear layer; OPL, outer plexiform layer; ONL, outer nuclear layer; ELM, external limiting membrane; IS-EZ, inner segment ellipsoid zone; IZ, interdigitation zone; RPE, retinal pigment epithelium. (TIF)

**S1 Table. Raw data for C57BL/6J mice.** Raw data for the retinal layer analysis (μm) in C57BL/6J mice. (PDF)

**S2 Table. Raw data for *Rdh5$^{-/-}$* mice.** Raw data for the retinal layer analysis (μm) in *Rdh5$^{-/-}$* mice. (PDF)

## Acknowledgments

The authors thank Dr. Brian Quinn for editing the English language of this manuscript.

## Author Contributions

**Data curation:** Takayuki Gonome, Kodai Yamauchi, Natsuki Maeda-Monai, Reiko Tanabu, Sei-ichi Ishiguro, Mitsuru Nakazawa.

**Formal analysis:** Yuting Xie, Sei-ichi Ishiguro.

**Funding acquisition:** Mitsuru Nakazawa.

**Investigation:** Yuting Xie, Takayuki Gonome, Kodai Yamauchi, Natsuki Maeda-Monai, Reiko Tanabu, Sei-ichi Ishiguro, Mitsuru Nakazawa.

**Methodology:** Sei-ichi Ishiguro, Mitsuru Nakazawa.

**Project administration:** Mitsuru Nakazawa.

**Resources:** Mitsuru Nakazawa.

**Supervision:** Sei-ichi Ishiguro, Mitsuru Nakazawa.

**Validation:** Sei-ichi Ishiguro.

**Visualization:** Yuting Xie.

**Writing – original draft:** Yuting Xie.

**Writing – review & editing:** Takayuki Gonome, Sei-ichi Ishiguro, Mitsuru Nakazawa.

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
