## [Decision Letter · Decision Letter 0]

5 Dec 2019

PONE-D-19-30197

A Spectral-Domain Optical Coherence Tomographic Analysis of RDH5-/- Mice Retina

PLOS ONE

Dear Professor Nakazawa,

Thank you for submitting your manuscript to PLOS ONE. After careful consideration, we feel that it has merit but does not fully meet PLOS ONE’s publication criteria as it currently stands. Therefore, we invite you to submit a revised version of the manuscript that addresses the points raised during the review process.

Please pay careful attention to the reviewers comments.

We would appreciate receiving your revised manuscript by Jan 19 2020 11:59PM. To enhance the reproducibility of your results, we recommend that if applicable you deposit your laboratory protocols in protocols.io, where a protocol can be assigned its own identifier (DOI) such that it can be cited independently in the future. For instructions see: http://journals.plos.org/plosone/s/submission-guidelines#loc-laboratory-protocols

We look forward to receiving your revised manuscript.

Kind regards,

Knut Stieger, D.V.M. Ph.D.

Academic Editor

PLOS ONE

Journal Requirements:

2. We noticed you have some minor occurrence of overlapping text with the following previous work, which needs to be addressed:

https://journals.plos.org/plosone/article?id=10.1371/journal.pone.0210439

In your revision ensure you cite all your sources (including your own works), and quote or rephrase any duplicated text outside the methods section. In the Methods, if you feel it necessary to repeat published protocol details, please rephrase if possible, indicate clearly that you are reproducing published information (e.g. “As described in detail previously [ref],…), and cite the relevant sources. Further consideration is dependent on these concerns being addressed."

3. Thank you for including your ethics statement: All experimental procedures performed in this study conformed to the regulations of the Association for Research in Vision and Ophthalmology (ARVO) Statement for the Use of Animals in Ophthalmic and Vision Research and were reviewed and approved by the Institutional Committee of Ethics for animal experiments (Approval Number: M12023).

Please amend your current ethics statement to include the full name of the ethics committee that approved your specific study.

For additional information about PLOS ONE submissions requirements for ethics oversight of animal work, please refer to http://journals.plos.org/plosone/s/submission-guidelines#loc-animal-research  

Reviewers' comments:

Reviewer's Responses to Questions

**Comments to the Author**

1. Is the manuscript technically sound, and do the data support the conclusions?

Reviewer #1: Yes

Reviewer #2: Partly

2. Has the statistical analysis been performed appropriately and rigorously? 

Reviewer #1: Yes

Reviewer #2: No

3. Have the authors made all data underlying the findings in their manuscript fully available?

Reviewer #1: Yes

Reviewer #2: Yes

4. Is the manuscript presented in an intelligible fashion and written in standard English?

Reviewer #1: Yes

Reviewer #2: Yes

5. Review Comments to the Author

Reviewer #1: In this manuscript entitled “A Spectral-Domain Optical Coherence Tomographic Analysis of RDH5-/- Mice Retina” Misuru Nakazawa and colleagues have described a detailed morphological phenotype of a RDH5-/- deficient mouse model for fundus albipunctatus (FA), a human hereditary retinal dystrophy. Based on the first description of the mouse mutant for the 11-cis-retinol dehydrogenase gene (11-cis-RoDH-/- mouse) by Driessen et al., 2000, the authors have investigated the retinal phenotype by using non-invasive diagnostic tools, e.g., OCT, SLO and ERG. Results were confirmed by histological and statistical analyses.

This manuscript offers a broad spectrum of retinal techniques for the investigation of a mild phenotype. The study is related to great diligence of work and is addressed to several interesting aspects in using this model for an improved understanding of FA. The authors have discussed important findings:

1) First description of a detailed morphological data set which reveal a mild phenotype caused by the gene disruption in RDH5.

2) Due to the morphological mild phenotype (decrease in longitudinal in the thickness of the retinal layers), the ERG recordings of mutants are inconspicuously compared to wild-type controls.

3) The whitish spots which are revealed in fundus images can not be depicted in any OCT-scans caused by limitation of the OCT-technique.

Comments and suggested improvements:

Style:

The notation of protein and gene should name according to the guidelines set by the International Committee on Standardized Genetic Nomenclature for Mice (https://www.jax.org/jax-mice-and-services/customer-support/technical-support/genetics-and-nomenclature) both for C57BL6 (e.g. C57BL/6N) and for the mutant.

In the discussion part the sentence “This may be due to spices-related differences...” should correct to “This may be due to species-related differences...”

Results:

Fig 2: The wavelength and the orientation of the OCT-scan (horizontal or vertical orientation) should be completed in the legend and/or in the images. The scans show predominantly a wavelike retinal structure. Is this phenomenon based on properties of the device or may due to any phenotype characteristics?

Fig 3: Please correct the arrow color in C.

Fig 5: Please adapt the position of the scale bars as well the size of images on a consistent standard.

Fig 7: Please adapt and correct the magnification properties in panel A-E. It appears that panel C shows a higher magnification. Panel F is not included in the Fig but described in the legend. Please correct.

Fig 8: It might be helpful for a better understanding of the ERG recordings if the months of investigation and the species (mutant vs control) are included into the figures.

Results and discussion:

The authors have declared no statistical significant differences in the ERG recordings between mutants and controls at PM3 and 5 (Fig 8). But, it might be that the b-wave part in mutations is smaller than in the controls at both time points of investigations by a careful consideration. Therefore, I recommend an overlay of the ERG recordings to proof this suggestion. Furthermore, I recommend to present a separate illustration of the ERG recordings under scotopic (dark adapted), rod-only, as well under scotopic (light adapted), cone-dominated, conditions (please see the publication: Tanimoto et al. 2009, Vision tests in the mouse: Functional phenotyping with electroretinography. Front Biosci (Landmark Ed). Additionally, in 2017, Kuehlewein et al. has described a cone dysfunction in patients with FA.

Reviewer #2: The authors investigated retinal structure and function in RDH5-/- mice. In general, it is a very good work, except for one topic (see below).

There are some remarks the authors should take into the consideration.

M&M:

The authors called the layers of photoreceptor inner segments and outer segments “photoreceptor layer C”, although the photoreceptor cell bodies stretch to the outer rim of the outer plexiform layer, with their nuclei constituting the outer nuclear layer. Maybe the authors could call their layer C “photoreceptor segments layer” or so?

Results:

The authors write: “No qualitative differences were observed in the retina between RDH5-/- and C57BL6 mice. In addition, this tendency was consistent even in the midperipheral areas (Fig 3).” Do they mean Fig. 2 instead of Fig. 3?

The reviewer has the impression that OCT scans shown in Figs. 3B and 3D do not correspond to the horizontal lines visible in Fig. 3A and 3C. Moreover, it would be nice to have arrows indicating the sites of the hyperreflective spots also in Figs. 3B and 3D.

In Figure 5, the scale bars are strange, and also the general appearance of the pictures. Were the pictures of the histological sections collected from different sources?

Discussion:

In the sentence “This may be due to spices-related difference between mice and humans.” May be a typo.

The biggest problem the reviewer sees in the presented manuscript are the electroretinography (ERG) data that are provided in supplement S4. Firstly, the number of animals (n=4) is by far too small to make any meaningful statements. In general, it is estimated that there may occur a deviation of 10-15% between single ERG measurements. In order to achieve sufficient statistical power to check for differences between experimental groups, approximately 10 animals per group are necessary.

As the next, it is very questionable to perform a normality test with a group of only n=4. It simply does not make any sense! Anyhow, Shapiro-Wilk normality test is not recommended in many cases, as the D'Agostino & Pearson omnibus normality test is better.

The deviations inside each group are very big, which raises the question if the ERG measurements were performed properly, and if these ERG data are useful at all. The reviewer has a long history in ERG and VEP measurements in rats and mice, and he finds the big deviations inside the groups very surprising, in particular with respect to the first mouse in the two PM5 groups. Moreover, the big decline of amplitudes in wild-type mice from PM3 to PM5 is surprising and was never observed by the reviewer in his own work. In contrast, almost no amplitude decline was seen in the RDH5-/- mice, which is suprising.

Consequently, the authors are advised to thouroughly check the ERG data and perfom the measurements in a higher number of animals, or to not touch the topic of ERG in their manuscript at all.

6. PLOS authors have the option to publish the peer review history of their article (what does this mean?). If published, this will include your full peer review and any attached files.

Reviewer #1: No

Reviewer #2: No

---

## [Author Response · Author response to Decision Letter 0]

16 Dec 2019

Responses to Reviewers’ comments

Reviewer #1

1. Style:

The notation of protein and gene should name according to the guidelines set by the International Committee on Standardized Genetic Nomenclature for Mice (https://www.jax.org/jax-mice-and-services/customer-support/technical-support/genetics-and-nomenclature) both for C57BL6 (e.g. C57BL/6N) and for the mutant.

Responses

We have changed the notation of protein and gene according to the guidelines set by the International Committee on Standardized Genetic Nomenclature for Mice. “RDH5-/- mice” has been changed to “Rdh5-/- mice” and “C57BL6 mice” has been changed to “C57BL/6J mice”, respectively. In addition, we have added the strain ID for each mouse group like “C57BL/6J (B6)” and “Rdh5-/- (B6;129S-Rdh5tm1Drie/J)” as recommended by the Committee in the Materials and methods section, Experimental subsection in page 6.

2. In the discussion part the sentence “This may be due to spices-related differences...” should correct to “This may be due to species-related differences...”

Responses

We have corrected the sentence to “This may be due to species-related differences...” in the Discussion section in page 13.

3. Results:

Fig 2: The wavelength and the orientation of the OCT-scan (horizontal or vertical orientation) should be completed in the legend and/or in the images. The scans show predominantly a wavelike retinal structure. Is this phenomenon based on properties of the device or may due to any phenotype characteristics?

Responses

The orientation of the OCT images has been added as “The position of the retinal SD-OCT image was set circumferentially around the optic disc (360°; diameter, 500 µm; 140 µm away from the optic disc margin, indicated by circles in Fig 1)” in Fig 2 Legend. The wavelike features in some OCT images were artificially made by the different angle between the mouse eye and the eye lens of the OCT apparatus. Therefore, we have added the sentence “The wavelike features in some images were artificially created depended on the angle of the mouse eye against the eye lens” in the Fig 2 Legend. The wavelength of the OCT-scan has been added in the Fig 2 Legend like “The SD-OCT images were made using light stimulation at 830 nm” in page 24. In addition, we have added the sentences “The wavelike features in some OCT images (Fig 2) were artificially created dependent on the angle between the mouse eye and the eye lens attached to the apparatus. Therefore, this phenomenon was not due to any of phenotypic characteristics but was based on properties of the device.” in the Results section in page 10.

4. Fig 3: Please correct the arrow color in C.

Responses

We have changed the arrow color in Fig A and C in black.

5. Fig 5: Please adapt the position of the scale bars as well the size of images on a consistent standard.

Responses

We have changed the position of scales in Fig 5.

6. Fig 7: Please adapt and correct the magnification properties in panel A-E. It appears that panel C shows a higher magnification. Panel F is not included in the Fig but described in the legend. Please correct.

Responses

We have added the explanation of the size of scale bars in the Fig 7 Legend as “Bars indicate 10 µm in panels A, B, D and E, and 2 µm in panel C”.

7. Fig 8: It might be helpful for a better understanding of the ERG recordings if the months of investigation and the species (mutant vs control) are included into the figures.

Results and discussion:

The authors have declared no statistical significant differences in the ERG recordings between mutants and controls at PM3 and 5 (Fig 8). But, it might be that the b-wave part in mutations is smaller than in the controls at both time points of investigations by a careful consideration. Therefore, I recommend an overlay of the ERG recordings to proof this suggestion. Furthermore, I recommend to present a separate illustration of the ERG recordings under scotopic (dark adapted), rod-only, as well under scotopic (light adapted), cone-dominated, conditions (please see the publication: Tanimoto et al. 2009, Vision tests in the mouse: Functional phenotyping with electroretinography. Front Biosci (Landmark Ed). Additionally, in 2017, Kuehlewein et al. has described a cone dysfunction in patients with FA.

Responses

As was also pointed out by the reviewer #2, we have recognized that our ERG data were inadequate, because of insufficient number of animals and only a single maximal rod-cone response without any of the other stimulation protocols. Because we are certain that the SD-OCT findings and morphological analyses of the retina in Rdh5-/- mice performed in this study are still valuable even without ERG analysis, we took the reviewer #2’s recommendation into account and we have decided that we do not touch the topic of ERG in the manuscript. Therefore, we have deleted the descriptions regarding ERG experiments including Fig 8 from the revised manuscript.

Reviewer #2

1. M&M:

The authors called the layers of photoreceptor inner segments and outer segments “photoreceptor layer C”, although the photoreceptor cell bodies stretch to the outer rim of the outer plexiform layer, with their nuclei constituting the outer nuclear layer. Maybe the authors could call their layer C “photoreceptor segments layer” or so?

Responses

We have changed the name “photoreceptor layer C” to “photoreceptor segments layer C” throughout the manuscript.

2. Results:

The authors write: “No qualitative differences were observed in the retina between RDH5-/- and C57BL6 mice. In addition, this tendency was consistent even in the midperipheral areas (Fig 3).” Do they mean Fig. 2 instead of Fig. 3?

Responses

The OCT images in Fig 2 are basically retinal section around the optic disc. Therefore, they do not reflect the findings of the mid-peripheral portion. The OCT images in Figs 3B and 3D partly include the retinal finding in the mid-peripheral portion. Therefore, we mention “No qualitative differences were observed in the retina between Rdh5-/- and C57BL/6J mice. In addition, this tendency was consistent even in the midperipheral areas (Fig 3).” In order not to make readers confused, we have changed these sentences to “No qualitative differences were observed in the retina between Rdh5-/- and C57BL/6J mice (Fig 2). In addition, this tendency was consistent even in the mid-peripheral areas (Figs 3B and 3D).” in page 10.

3. The reviewer has the impression that OCT scans shown in Figs. 3B and 3D do not correspond to the horizontal lines visible in Fig. 3A and 3C. Moreover, it would be nice to have arrows indicating the sites of the hyperreflective spots also in Figs. 3B and 3D.

Responses

Figs 3B and 3D do correspond to the horizontal lines, but orientations are opposite to those of Figs 3A and 3C. This derives from the SD-OCT machine’s property. To make the putative readers correctly understand this relationship, we have added the sentence “Note that the orientations of B and D are opposite to those of A and C.” in Fig 3 Legend. The arrows indicate the whitish spots in the fundus photos not the hyperreflective spots. Because we were unable to detect any hyperreflective spots corresponding to the whitish spots, we do not think that we need to add arrows at the “hyperreflective spots”.

4. In Figure 5, the scale bars are strange, and also the general appearance of the pictures. Were the pictures of the histological sections collected from different sources?

Response

As pointed out by both reviewers, we have corrected the position of scale bars.

5. Discussion:

In the sentence “This may be due to spices-related difference between mice and humans.” May be a typo.

Responses

As pointed out by both reviewers, we have corrected the typographical error.

6. The biggest problem the reviewer sees in the presented manuscript are the electroretinography (ERG) data that are provided in supplement S4. Firstly, the number of animals (n=4) is by far too small to make any meaningful statements. In general, it is estimated that there may occur a deviation of 10-15% between single ERG measurements. In order to achieve sufficient statistical power to check for differences between experimental groups, approximately 10 animals per group are necessary.

As the next, it is very questionable to perform a normality test with a group of only n=4. It simply does not make any sense! Anyhow, Shapiro-Wilk normality test is not recommended in many cases, as the D'Agostino & Pearson omnibus normality test is better.

The deviations inside each group are very big, which raises the question if the ERG measurements were performed properly, and if these ERG data are useful at all. The reviewer has a long history in ERG and VEP measurements in rats and mice, and he finds the big deviations inside the groups very surprising, in particular with respect to the first mouse in the two PM5 groups. Moreover, the big decline of amplitudes in wild-type mice from PM3 to PM5 is surprising and was never observed by the reviewer in his own work. In contrast, almost no amplitude decline was seen in the RDH5-/- mice, which is surprising.

Consequently, the authors are advised to thoroughly check the ERG data and perform the measurements in a higher number of animals, or to not touch the topic of ERG in their manuscript at all.

Responses

As pointed out by both reviewers, we have realized that the ERG data in this study were quite insufficient because of the comments raised by the reviewer #2. Because it will be difficult for us to further perform ERG assessments so as to fulfill the reviewers’ comments and because we believe that our OCT data with morphological analyses would be still valuable even without ERG analyses, we have decided not to touch the topic of ERG in this manuscript at all, according to the reviewer’s suggestion. Therefore, we have deleted the portions describing ERG from the revised manuscript. In addition to the Shapiro-Wilk test, we also carried out the Kolmogorov-Smirnov test to check the normality of the data.

---

## [Decision Letter · Decision Letter 1]

14 Feb 2020

PONE-D-19-30197R1

A Spectral-Domain Optical Coherence Tomographic Analysis of Rdh5-/- Mice Retina

PLOS ONE

Dear Professor Nakazawa,

Thank you for submitting your manuscript to PLOS ONE. After careful consideration, we feel that it has merit but does not fully meet PLOS ONE’s publication criteria as it currently stands. Therefore, we invite you to submit a revised version of the manuscript that addresses the points raised during the review process.

Please address the final points raised by reviewer 1. Furthermore, please elaborate more on the point that without the functional readout (ie. ERG), the dataset contains sufficient amount of new data to be published in PONE. 

We would appreciate receiving your revised manuscript by Mar 30 2020 11:59PM. To enhance the reproducibility of your results, we recommend that if applicable you deposit your laboratory protocols in protocols.io, where a protocol can be assigned its own identifier (DOI) such that it can be cited independently in the future. For instructions see: http://journals.plos.org/plosone/s/submission-guidelines#loc-laboratory-protocols

We look forward to receiving your revised manuscript.

Kind regards,

Knut Stieger, D.V.M. Ph.D.

Academic Editor

PLOS ONE

Reviewers' comments:

Reviewer's Responses to Questions

**Comments to the Author**

1. If the authors have adequately addressed your comments raised in a previous round of review and you feel that this manuscript is now acceptable for publication, you may indicate that here to bypass the “Comments to the Author” section, enter your conflict of interest statement in the “Confidential to Editor” section, and submit your "Accept" recommendation.

Reviewer #1: All comments have been addressed

Reviewer #2: All comments have been addressed

2. Is the manuscript technically sound, and do the data support the conclusions?

Reviewer #1: Partly

Reviewer #2: Yes

3. Has the statistical analysis been performed appropriately and rigorously? 

Reviewer #1: Yes

Reviewer #2: Yes

4. Have the authors made all data underlying the findings in their manuscript fully available?

Reviewer #1: Yes

Reviewer #2: Yes

5. Is the manuscript presented in an intelligible fashion and written in standard English?

Reviewer #1: Yes

Reviewer #2: Yes

6. Review Comments to the Author

Reviewer #1: Misuru Nakazawa and colleagues have revised the manuscript entitled “A Spectral-Domain Optical Coherence Tomographic Analysis of RDH5-/- Mice Retina” by following the comments of both reviewers. The major revision which was made by the authors was the deletion of the obtained ERG data at all. The rationale for that decision was the awareness that the ERG data are insufficiently in terms of animal numbers and protocols used for the experiments to allow any interpretation of the RDH5 murine mutant functional phenotype.

Comments:

For two requested changes for the following aspects, the edit could not found in the revised version:

1) In the discussion part the sentence “This may be due to spices-related differences...” should correct to “This may be due to species-related differences...”

2) Fig 3: Please correct the arrow colour in C.

Reviewer #2: The comments of the reviewer have been addressed. It is good that the topic of electroretinography was omitted as long as there are no valid data available.

7. PLOS authors have the option to publish the peer review history of their article (what does this mean?). If published, this will include your full peer review and any attached files.

Reviewer #1: No

Reviewer #2: No

---

## [Author Response · Author response to Decision Letter 1]

19 Feb 2020

Responses to Reviewers’ comments

Reviewer #1

1. In the discussion part the sentence “This may be due to spices-related differences...” should correct to “This may be due to species-related differences...”

Response:

Thank you very much for detecting the typographical error. We corrected the miss-spelled word “spices” to “species” in the Discussion section.

2. Fig 3: Please correct the arrow colour in C.

Response:

As suggested by the reviewer, we changed the color of the horizontal arrow (line) from green to black in Fig 3.

---

## [Editor Report · Decision Letter 2]

26 Feb 2020

PONE-D-19-30197R2

A Spectral-Domain Optical Coherence Tomographic Analysis of Rdh5-/- Mice Retina

PLOS ONE

Dear Professor Nakazawa,

Thank you for submitting your manuscript to PLOS ONE. After careful consideration, we feel that it has merit but does not fully meet PLOS ONE’s publication criteria as it currently stands. Therefore, we invite you to submit a revised version of the manuscript that addresses the points raised during the review process.

Dear Authors, 

you did not address my point on elaborating on the issue of why do you think that a pure morphological study describing the retinal changes without discussing the potential effect on the retinal function is sufficient for publication in PONE. I would like to encourage you to do so.

We would appreciate receiving your revised manuscript by Apr 11 2020 11:59PM. To enhance the reproducibility of your results, we recommend that if applicable you deposit your laboratory protocols in protocols.io, where a protocol can be assigned its own identifier (DOI) such that it can be cited independently in the future. For instructions see: http://journals.plos.org/plosone/s/submission-guidelines#loc-laboratory-protocols

We look forward to receiving your revised manuscript.

Kind regards,

Knut Stieger, D.V.M. Ph.D.

Academic Editor

PLOS ONE

---

## [Author Response · Author response to Decision Letter 2]

7 Mar 2020

Responses to the editor’s comments

Thank you for your valuable comments.

“Why do you think that a pure morphological study describing the retinal changes without discussing the potential effect on the retinal function is sufficient for publication in PONE?”

Response:

Previously, Driessen et al. [16] and Kim et al. [19] have extensively performed ERG experiments in detail. Their results are summarized by the fact that the photoexcitation of the a- and b-waves was not defective in RDH5-/- mice using standard dark-adapted flash ERG, but the recovery of the a-wave from an intensive bleaching condition was delayed in RDH5-/- mice compared with that in wild-type mice. Because their experiments were quite extensively and delicately performed, we believe that their results are quite informative and that we do not need to add any further ERG experiments to be performed in this study. The previous reports have detected substantially mild ERG defects in RDH5-/- mice [16, 19]. We speculate that these mild functional defects may be related with the relatively mild structural changes observed in this study using SD-OCT and electron microscopy. Particularly, we conclude that SD-OCT can quantitatively and non-invasively analyze morphologic changes.

For these reasons, we have changed the paragraph in the Discussion section (P.14) of the previous manuscript, “A study of the photoreceptor function in Rdh5-/-Rdh11-/- mice conducted by Maeda et al. showed that aberrant cone responses were detected until 12-months of age by flicker ERG [41]. Therefore, we speculate that these differences in the metabolisms between mice and humans may be why abnormal findings were hardly detected, at least qualitatively, in the SD-OCT images of Rdh5-/- mice. In addition, we consider that these metabolic differences between mice and humans may lead to the lack of any marked differences in the amplitudes of ERG a- and b-waves in both Rdh5-/- and C57BL/6J mice [16, 19] “ 

to

 “Regarding functional aspects, since Driessen et al. [16] and Kim et al. [19] have previously extensively performed electrophysiologic studies in detail, we thought that the results of their studies were quite informative, even though we did not perform electroretinographic (ERG) experiments in this study. Their studies regarding the photoreceptor function in RDH5-/- mice can be summarized by the fact that the photoexcitation function was not defective in RDH5-/- mice using standard dark-adapted single flash ERG experiments [16, 19]. However, the recovery of the a-wave from an intensive bleaching condition was delayed in RDH5-/- mice compared with that in wild-type mice, although no histologic difference was detected between RDH5-/- and wild-type mice [16, 19]. The results of these previous studies suggested that RDH5-/- mice showed substantially milder electrophysiological deficits than patients with FA. We speculate that these differences in the electrophysiology may be due to the different metabolisms between mice and humans. In addition, these differences in the metabolisms between mice and humans may be why hardly any abnormal findings were detected, at least qualitatively, in the SD-OCT images of Rdh5-/- mice. In the revised manuscript (P14, L267 to P15, L281).

In addition, we have added the sentences “However, it is possible that these quantitative changes observed by SD-OCT may be related to the defective photo-recovery function in RDH5-/- mice [16, 19]. Further studies should be carried out in the future to clarify this point.” in P 15, L280-282 to clarify the ERG findings previously reported.

---

## [Editor Report · Decision Letter 3]

19 Mar 2020

A Spectral-Domain Optical Coherence Tomographic Analysis of Rdh5-/- Mice Retina

PONE-D-19-30197R3

Dear Dr. Nakazawa,

We are pleased to inform you that your manuscript has been judged scientifically suitable for publication and will be formally accepted for publication once it complies with all outstanding technical requirements.

With kind regards,

Knut Stieger, D.V.M. Ph.D.

Academic Editor

PLOS ONE
---

## [Editor Report · Acceptance letter]

23 Mar 2020

PONE-D-19-30197R3 

A Spectral-Domain Optical Coherence Tomographic Analysis of *Rdh5^-/-^* Mice Retina

Dear Dr. Nakazawa:

I am pleased to inform you that your manuscript has been deemed suitable for publication in PLOS ONE. Congratulations! Your manuscript is now with our production department. 

With kind regards,

on behalf of

Dr. Knut Stieger 

Academic Editor

PLOS ONE